# Design, Synthesis, and Biological Evaluation of Pyridineamide Derivatives Containing a 1,2,3-Triazole Fragment as Type II c-Met Inhibitors

**DOI:** 10.3390/molecules25010010

**Published:** 2019-12-18

**Authors:** Hehua Xiong, Jianxin Cheng, Jianqing Zhang, Qian Zhang, Zhen Xiao, Han Zhang, Qidong Tang, Pengwu Zheng

**Affiliations:** School of Pharmacy, Jiangxi Science & Technology Normal University, Nanchang 330013, China; 18296154955@163.com (H.X.); cjx3159@163.com (J.C.); zhang1405474771@163.com (J.Z.); 15797966937@163.com (Q.Z.); xz950420@163.com (Z.X.); zbl1045762244@163.com (H.Z.)

**Keywords:** 4-(pyridin-4-yloxy)benzamide, 1,2,3-triazole, c-Met, inhibitor

## Abstract

A series of 4-(pyridin-4-yloxy)benzamide derivatives containing a 1,2,3-triazole fragment were designed, synthesized, and their inhibitory activity against A549, HeLa, and MCF-7 cancer cell lines was evaluated. Most compounds exhibited moderate to potent antitumor activity against the three cell lines. Among them, the promising compound **B26** showed stronger inhibitory activity than Golvatinib, with IC_50_ values of 3.22, 4.33, and 5.82 μM against A549, HeLa, and MCF-7 cell lines, respectively. The structure–activity relationships (SARs) demonstrated that the modification of the terminal benzene ring with a single electron-withdrawing substituent (fluorine atom) and the introduction of a pyridine amide chain with a strong hydrophilic group (morpholine) to the hinge region greatly improved the antitumor activity. Meanwhile, the optimal compound **B26** showed potent biological activity in some pharmacological experiments in vitro, such as cell morphology study, dose-dependent test, kinase activity assay, and cell cycle experiment. Finally, the molecular docking simulation was performed to further explore the binding mode of compound **B26** with c-Met.

## 1. Introduction

Cancer has become a serious threat to human life and health [1,2]. c-Met tyrosine kinase is a kind of type III tyrosine kinase, which is closely related to cellular activities [3] such as growth, reproduction, metastasis, etc. However, abnormal expression of c-Met kinases in cells usually triggers the occurrence, invasion, and metastasis of various cancer diseases [4]. c-Met kinase has been found overexpressed in various cancer cells and has become an attractive target for cancer treatment [5,6]. Nowadays, developing small molecule c-Met kinase inhibitors has become a hotspot in the treatment of human cancer [7].

c-Met inhibitors are classified as type I and type II inhibitors according to the binding mode of inhibitors with c-Met. Usually, c-Met inhibitors of type I are single-target inhibitors that bind to the hinge region of the ATP pocket, and c-Met inhibitors of type II are multitarget inhibitors that bind to the hinge region and an extra hydrophobic pocket [8,9]. With the development of biology and pharmacology, the intracellular mechanisms have been elucidated, and research in small molecule inhibitors has made significant progress. Type II c-Met inhibitors possess more potent inhibitory activity and better tolerance to drug resistance through binding to kinases of multiple types. Cabozantinib, a c-Met inhibitor of type II, was approved by the FDA for treatment of prostate cancer in 2012. Some representative c-Met inhibitors (type II) in the clinical trial phase are listed in Figure 1, including Golvatinib, BMS-777607, Altriatinib, and TAS-115 [10,11,12]. Furthermore, according to the structure characteristics of inhibitors, the skeleton of c-Met inhibitor (type II) was summarized into blocks A, B, C, and D by our research group, as shown in Figure 1 [13].

We explored the SARs of c-Met inhibitors (type II) bearing pyridinamide structure using Golvatinib as a lead compound in this study. In order to guide our modification of the lead compound, a molecular docking simulation of Golvatinib and c-Met protein was performed, as shown in Figure 2. The docking results indicated that blocks A and C, which possessed hydrogen bonding interactions with residues Met1160, Asp1222, and Lys1110, played a key role in maintaining the inhibitory activity against c-Met. Therefore, the modification was mainly concentrated on blocks A and C. Firstly, a pyridylamide structure in block A, including hydrogen bond donor and hydrogen bond acceptor atoms, was introduced into the meta position of pyridine in order to enhance the interaction with the hinge region residues, and then some alkyl chains bearing morpholine or thiophene groups were connected to the terminal of the pyridine amide structure for better water solubility of compounds. Then, a 1,2,3-triazole fragment with good biological activity and two hydrogen bond acceptor atoms was introduced to block C for more tight combination between the compounds and c-Met. In addition, considering the steric hindrance problem [14], H/F atoms and different substituents were introduced in blocks B and D, respectively, to explore their effects on biological activity of target compounds. Finally, a series of small molecule inhibitors containing the 1,2,3-triazole fragment were designed, synthesized, and biologically evaluated.

## 2. Results and Discussion

### 2.1. Chemistry

The synthesis route of target compounds **B1–B27** is shown in Scheme 1. Firstly, the commercially available pyridinecarboxylic acid **6** was reacted with SOCl_2_ using NaBr as catalyst to yield 4-chloropicolinoyl chloride **7**, which was then converted into ethyl 4-chloropicolinate **8** with ethanol. Next, 4-chloropicolinate **8** was subjected to nucleophilic substitute with *p*-nitrophenol or 2-fluoro-4-nitrophenol using chlorobenzene as solvent to give picolinate analogues **9a–9b**, which were then dissolved in 1,4-dioxane and hydrolyzed with NaOH aqueous solution to obtain picolinic acid analogues **10a–10b**. Intermediates **10a–10b** were reacted with SOCl_2_ to get the corresponding acyl chlorides and then connected with amines (including propan-1-amine, 2-(thiophen-2-yl)ethan-1-amine, pyrrolidine and 3-morpholinopropan-1-amine) to obtain picolinamide analogues **11a–11e**. Finally, the key intermediates **12a–12e** were produced by the reduction of **11a–11e** with hydrazine hydrate as reducing agent. Other intermediates **13a–13i**, which have been reported in our previous research [15], were combined with intermediates **12a**–**12e** by a nucleophilic substitute reaction to get the target compounds **B1–B27**.

The structural information of the target compounds was confirmed by ^1^H-NMR, ^13^C-NMR, and TOF MS (ES+), which were consistent with the structures depicted.

### 2.2. Antitumor Activity of Compounds **B1–B27** Against A549, HeLa, and MCF-7 Cell Lines and SAR Analysis

The antitumor activity of target compounds against A549, HeLa, and MCF-7 cell lines was evaluated to investigate their inhibitory activity against cancer cells [16]. The antitumor activities of compounds were displayed as the IC_50_ (half-inhibitory concentration) values, as shown in Table 1 and Table 2. Most compounds exhibited moderate to potent antitumor activity against three cell lines. Compounds **B25–B27** possessed more potent inhibitory activity than Golvatinib, as shown in Table 2, with IC_50_ values in the range of 3.22–13.60 μM. The optimal compound **B26** showed excellent inhibitory activity against A549, HeLa, and MCF-7 cell lines with IC_50_ values of 3.22 ± 0.12, 4.33 ± 0.09, and 5.82 ± 0.09 μM, respectively, which were more efficient than Golvatinib (8.14 ± 0.45, 15.17 ± 0.17, and 16.91 ± 0.29 μM).

According to the antitumor activity of compounds **B1–B27** listed in Table 1 and Table 2, the SARs were summarized as follows. Overall, the F atom on the meta-central benzene ring provided stronger inhibitory activity against cancer cells than H atom, for example, compounds **B1–B8** displayed less inhibitory activity than compounds **B9–B18**. A single substitution (such as H/F/OCF_3_/Cl/CF_3_) linked to the terminal benzene ring was favorable for maintaining the biological activity, such as compounds **B1–B5** and **B10–B15** that possessed moderate antitumor activity with IC_50_ values less than 60 μM against A549 and HeLa cell lines. However, the double substituents connected to the terminal benzene ring obviously impaired the inhibitory activity of compounds **B6–B9** and **B15–B18,** whose IC_50_ values were more than 120 μM against three cell lines. A possible explanation would be that the introduction of two substituents to the terminal benzene ring may create steric hindrance, which made it difficult for compounds to extend into the hydrophobic pocket and to bind tightly to c-Met kinase.

Inspired by the inhibitory activity of compounds **B1–B3** and **B10–B12**, as shown in Table 1, the F atom of the central benzene ring and the single group (H, F, or OCF_3_) of the terminal benzene ring were reserved to perform a SAR study of the pyridine amide moiety. Hydrophilic groups, including morpholine or thiophene groups attached to the terminal of pyridine amide, have a great contribution to inhibitory activity of the compounds **B22–B27** (with IC_50_ values in the range of 3.22–45.18 μM).

In summary, introducing morpholino groups and F atoms to the pyridylpropyl amide and the terminal benzene ring, respectively, made significant advances in the inhibitory activity.

### 2.3. Dose-Dependent Test of Compound **B26** Against A549 Cells

To explore the relationship between concentrations and antitumor activity of compound **B26**, we carried out a dose-dependent test based on the MTT method [16], in which A549 cells were treated with seven different concentrations of compound **B26** (0.137, 0.411, 1.234, 3.704, 11.11, 33.33, and 100 μM), using Golvatinib as a positive control, as seen in Figure 3. The results revealed that compound **B26** could effectively inhibit cancer cells in a dose-dependent manner. Notably, compound **B26** inhibited A549 cells more than 50% at the concentration of 3.704 μM, while the inhibitory rate of Golvatinib was less than 40% at the same concentration.

### 2.4. Cell Cycle Study of A549 Cells Treated with Compound **B26**

To further investigate the inhibitory pattern of compound **B26** on the growth of cancer cells, cell cycle distribution analysis [17] was carried out on A549 cells, as shown in Figure 4. Compound **B26** at concentrations of 5, 10, and 15 μM and Golvatinib at a concentration of 5 μM were delivered to A549 cells for 24 h, as shown in Figure 4. With the increased concentration of compound **B26**, the percentage of cells blocked in G_0_/G_1_ phase was increased from 65.44% to 79.18%, while the percentage in the S phase decreased from 21.91% to 16.56%. There was no significant change in G_2_/M phase. The result showed that compound **B26** (74.41%) and Golvatinib (74.77%) had the same inhibitory effect on A549 cells at the concentration of 5 μM, both of which could block cells in the G_0_/G_1_ phase.

### 2.5. c-Met Enzyme Assay of Compounds **B25–B27** and Staurosporine

Compounds **B25**, **B26**, and **B27**, possessing potent antitumor activity against A549, HeLa, and MCF-7 cell lines, were selected to investigate their c-Met kinase assay by mobility shift assay method [18] with staurosporine as a positive control, which was used to insure the reliability of the experimental data. Compound **B26** exhibited potent inhibitory activity against c-Met with an inhibition rate of 36% at the concentration of 0.625 μM, as shown in Table 3.

### 2.6. Molecular Docking Simulation of Compounds (**B26** and Golvatinib) and c-Met

Next, we further explored the binding mode of compound **B26** and Golvatinib with c-Met kinase (PDB: 3LQ8, extracted from the PDB database) through molecule docking simulation using AutoDock 4.2 software [19], as shown in Figure 5. The docking results indicated that compound **B26** and Golvatinib in an extended conformation formed strong hydrogen bonding interactions with residues Met1160, Asp1222, and Lys1110, as shown in Figure 5a–d. Interestingly, the pyridylamide structure of compound **B26** formed a bidentate hydrogen bond with residue Met1160, as shown in Figure 5a,b, which may offer a stronger ability of compound **B26** to bind with c-Met. Compound **B26** formed more hydrogen bonding interactions with Met1160 than Golvatinib, which may explain the better antitumor activity of compound **B26**.

## 3. Experimental Section

### 3.1. Chemistry

Unless specifically required, all reagents used in the experiments were purchased as commercial analytical grade and used without further purification. Frequently used solvents (dichloromethane, ethyl acetate, tetrahydrofuran, 1,4-dioxane, methanol, ethanol, etc.) were absolutely anhydrous. Flash column chromatography was performed on silica gel (300 mesh) using a mixture of petroleum ether (PE) and ethyl acetate (EA). The reaction process was monitored by thin-layer chromatography (TLC, SH-GH254) and spots were visualized with ultraviolet analyzer (light in 254 or 365 nm). ^1^H-NMR and ^13^C-NMR spectra analysis of compounds was implemented in Bruker 400 MHz spectrometer (Bruker Bioscience, Billerica, MA, USA) using tetramethylsilane (TMS) as an internal standard at room temperature. Mass spectrometry (MS) of target compounds was carried out Waters High Resolution Quadrupole Time of Flight Tandem Mass Spectrometry (Waters, Xevo G2-XS Tof). The melting point of the compounds was measured by SGW X-4 micro melting point instrument. The purity of compounds was determined by an Agilent 1260 liquid chromatograph equipped with an Inertex-C18 column, and all were more than 95%.

#### 3.1.1. Preparation of 4-Chloropicolinoyl Chloride (**7**)

Pyridinecarboxylic acid **6** (10.00 g, 0.081 mol), NaBr (0.10 g, 0.001 mol), and a drop of *N,N*-dimethylformamide (DMF) were added in thionyl chloride (SOCl_2_, 50 mL) and stirred at 85 °C for 18 h. The reaction was monitored by TLC. After completion of the reaction, the solvent was removed under vacuum to obtain a yellow liquid, which was dissolved in DCM for further use.

#### 3.1.2. Preparation of Ethyl 4-Chloropicolinate (**8**)

Dichloromethane (30 mL), ethanol (16.43 g, 0.170 mol), and triethylamine (17.17 g, 0.170 mol) were successively added to a beaker and stirred at 0 °C for 0.5 h. Subsequently, a mixture of 4-chloropicolinoyl chloride **7** (16.43 g, 0.110 mol) and DCM (5 mL) was added dropwise, stirred at 0 °C for 0.5 h, and monitored by TLC. After completion of the reaction, NaOH solution was added to the mixture to adjust the pH value 9–10. Ultimately, the organic layers were combined and dried to obtain a brown liquid.

#### 3.1.3. Preparation of Ethyl 4-(4-Nitrophenoxy)picolinate (**9a**) and Ethyl4-(2-Fluoro-4-Nitrophenoxy)picolinate (**9b**)

Intermediate **8** (13.22 g, 0.071 mol) and *p*-nitrophenol or 2-fluoro-4-nitrophenol (0.026 mol) were dissolved in chlorobenzene (40 mL) and then stirred at 130 °C for 4 h. The completion of the reaction was monitored by TLC. After completion of the reaction, the cold petroleum ether (100 mL) was poured into the reaction solvent and stirred at room temperature for 0.5 h. Next, petroleum ether (100 mL) was poured off to give a viscous liquid, which was then dissolved in DCM (300 mL) and extracted with aqueous solution of NaOH. The organic layer was combined, dried, and concentrated under vacuum to get a pale yellow solid.

#### 3.1.4. Preparation of 4-(4-Nitrophenoxy)picolinic Acid (**10a**) and 4-(2-Fluoro-4-Nitrophenoxy)picolinic Acid (**10b**)

Intermediates **9a–9b** (0.035 mol) were dissolved in 1,4-dioxane (50 mL) and stirred at room temperature for 0.5 h. A solution of 10% NaOH (2.28 mL, 0.057 mol) was then added dropwise to the above mixture, stirred at room temperature for 0.5 h, and monitored by TLC. After completion of the reaction, the solvent was evaporated under vacuum to give a white solid. The white solid was then dissolved in saturated NaCl solution (500 mL) and stirred at room temperature for 10 h. Subsequently, after the pH of the solution was adjusted to 2–3 by 75% hydrochloric acid (HCl), a yellow solid was precipitated, filtered, and dried.

#### 3.1.5. Preparation of 4-(4-Nitrophenoxy)picolinamide or 4-(2-Fluoro-4-Nitrophenoxy)picolinamide Analogues (**11a–11e**)

Intermediates **10a–10b** (0.020 mol) and one drop of DMF were successively dissolved in SOCl_2_ (50 mL) and stirred at 85 °C for 0.5 h. Then reaction solvent was concentrated in vacuum, dissolved in DCM (8 mL), and added dropwise to a mixture of DCM (50 mL), ammonia analogs (0.020 mol), and triethylamine (0.020 mol). The reaction solution was stirred at 0 °C for 0.5 h, and monitored by TLC. Finally, a gray slid was obtained by the same post-treatment method as that used for preparation of ethyl 4-chloropicolinate **8**.

#### 3.1.6. Preparation of 4-(4-Aminophenoxy)picolinamide or 4-(2-Fluoro-4-aminophenoxy)picolinamide Analogues (**12a–12e**)

Intermediates **11a–11e** (0.016 mol), activated carbon (1.92 g, 0.160 mol), and FeCl_3_·6H_2_O (0.43 g, 0.016 mol) were dissolved in ethanol (30 mL) and stirred at 90 °C for 0.5 h. Then, 80% hydrazine hydrate (8.00 g, 0.128 mol) was added dropwise and stirred for 3.5 h. The reaction was monitored by TLC. After completion of the reaction, the solution was filtered, concentrated, and recrystallized to give a green solid.

#### 3.1.7. Preparation of 5-Methyl-1-Phenyl-1H-1,2,3-Triazole-4-Carbonyl Chloride or 1-Phenyl-5-(trifluoromethyl)-1H-1,2,3-Triazole-4-Carbonyl Chloride Analogues (**13i–13e**)

The preparation of intermediates **13a–13i** was carried out according to our previous research [15].

#### 3.1.8. Preparation of Target Compounds **B1–B27**

Intermediates **12a–12e** (0.002 mol) and sodium bicarbonate (4.20 g, 0.005 mol) were dissolved in DCM and stirred at 0 °C for 0.5 h. The solution of intermediates **13a–13i** (0.002 mol) and DCM were added dropwise to the above mixture, and stirred at 0 °C for 0.5 h. The reaction was monitored by TLC. After completion of the reaction, the solution was extracted with a mixture of DCM/NaOH (500 mL) three times and then the organic layers were combined, dried over sodium sulfate, evaporated, and purified by chromatographic column to possess a light yellow or white solid.

*4-(4-(5-methyl-1-phenyl-1H-1,2,3-triazole-4-carboxamido)phenoxy)-N-propylpicolinamide* (**B1**) Light yellow solid; Yield: 78.5%; m.p.: 116.8–117.1 °C; ^1^H-NMR (400 MHz, DMSO-*d*_6_, ppm) δ 10.75 (s, 1H), 8.82 (s, 1H), 8.52 (d, *J* = 5.5 Hz, 1H), 8.03 (s, 1H), 8.01 (s, 1H), 7.67 (s, 2H), 7.64 (d, *J* = 6.4 Hz, 2H), 7.43 (s, 1H), 7.23 (d, *J* = 8.4 Hz, 2H), 7.18 (d, *J* = 4.5 Hz, 1H), 3.22 (dd, *J* = 12.8, 6.3 Hz, 2H), 2.59 (s, 3H), 1.50 (dt, *J* = 14.1, 7.0 Hz, 2H), 0.84 (t, *J* = 7.2 Hz, 3H). TOF MS ES+ (*m/z*): [M + H]^+^, calcd for C_25_H_24_N_6_O_3_: 457.1988, found, 457.1986.

*4-(4-(1-(4-fluorophenyl)-5-methyl-1H-1,2,3-triazole-4-carboxamido)phenoxy)-N-propylpicolinamide* (**B2**) Light yellow solid; Yield: 59.3%; m.p.: 110.4–110.8 °C; ^1^H-NMR (400 MHz, DMSO-*d*_6_, ppm) δ 10.64 (s, 1H), 8.71 (s, 1H), 8.40 (d, *J* = 5.6 Hz, 1H), 7.91 (s, 1H), 7.89 (s, 1H), 7.65 (d, *J* = 4.8 Hz, 1H), 7.63 (d, *J* = 4.6 Hz, 1H), 7.39 (t, *J* = 8.7 Hz, 2H), 7.31 (d, *J* = 2.1 Hz, 1H), 7.11 (d, *J* = 8.8 Hz, 2H), 7.06 (d, *J* = 3.1 Hz, 1H), 3.10 (dd, *J* = 13.3, 6.6 Hz, 2H), 2.46 (s, 3H), 1.38 (dt, *J* = 14.2, 7.1 Hz, 2H), 0.72 (t, *J* = 7.3 Hz, 3H). TOF MS ES+ (*m/z*): [M + H]^+^, calcd for C_25_H_23_FN_6_O_3_: 475.1894, found, 475.1896.

*4-(4-(5-methyl-1-(2-(trifluoromethoxy)phenyl)-1H-1,2,3-triazole-4-carboxamido)phenoxy)-N-propylpicolinamide* (**B3**) White solid; Yield: 78.2%; m.p.: 97.7–98.0 °C; ^1^H-NMR (400 MHz, DMSO-*d*_6_, ppm) δ 10.81 (s, 1H), 8.81 (s, 1H), 8.52 (d, *J* = 5.3 Hz, 1H), 8.03 (s, 1H), 8.01 (s, 1H), 7.88 (d, *J* = 7.7 Hz, 1H), 7.84 (d, *J* = 7.6 Hz, 1H), 7.78 (d, *J* = 7.7 Hz, 1H), 7.72 (t, *J* = 7.2 Hz, 1H), 7.42 (s, 1H), 7.23 (d, *J* = 8.6 Hz, 2H), 7.18 (d, *J* = 0.8 Hz, 1H), 3.22 (d, *J* = 6.2 Hz, 2H), 2.46 (s, 3H), 1.51 (dd, *J* = 14.0, 7.0 Hz, 2H), 0.84 (t, *J* = 7.2 Hz, 3H). TOF MS ES+ (*m/z*): [M + H]^+^, calcd for C_26_H_23_F_3_N_6_O_4_: 541.1811, found, 541.1813.

*4-(4-(1-(4-chlorophenyl)-5-(trifluoromethyl)-1H-1,2,3-triazole-4-carboxamido)phenoxy)-N-propylpicolinamide* (**B4**) White solid; Yield: 66.4%; m.p.: 159.0–159.2 °C; ^1^H-NMR (400 MHz, DMSO-*d*_6_, ppm) δ 11.19 (s, 1H), 8.80 (s, 1H), 8.52 (d, *J* = 5.5 Hz, 1H), 7.99 (s, 1H), 7.97 (d, *J* = 1.2 Hz, 1H), 7.79 (d, *J* = 4.4 Hz, 3H), 7.75 (d, *J* = 8.7 Hz, 1H), 7.42 (s, 1H), 7.27 (d, *J* = 8.7 Hz, 2H), 7.19 (d, *J* = 5.5 Hz, 1H), 3.23 (dd, *J* = 13.2, 6.7 Hz, 2H), 1.57–1.47 (m, 2H), 0.84 (t, *J* = 7.4 Hz, 3H). ^13^C-NMR (100 MHz, DMSO-*d*_6_, ppm) δ 165.73, 163.13, 156.58, 152.54, 150.36(2,C), 149.46, 142.31, 136.19, 135.84, 134.09, 129.70(2,C), 128.22(2,C), 122.40(2,C), 121.33(2,C), 114.18(2,C), 108.95, 40.58, 22.34, 11.27. TOF MS ES+ (*m/z*): [M + H]^+^, calcd for C_25_H_20_ClF_3_N_6_O_3_: 545.1316, found, 545.1317.

*N-propyl-4-(4-(5-(trifluoromethyl)-1-(2-(trifluoromethyl)phenyl)-1H-1,2,3-triazole-4-carboxamido)phenoxy)picolinamide* (**B5**) White yellow solid; Yield: 69.4%; m.p.: 163.4–163.7 °C; ^1^H-NMR (400 MHz, DMSO-*d*_6_, ppm) δ 11.26 (s, 1H), 8.80 (s, 1H), 8.53 (d, *J* = 5.2 Hz, 1H), 8.15–8.09 (m, 2H), 8.05 (d, *J* = 7.2 Hz, 1H), 8.00 (d, *J* = 7.7 Hz, 3H), 7.43 (s, 1H), 7.27 (d, *J* = 8.5 Hz, 2H), 7.18 (d, *J* = 2.7 Hz, 1H), 3.23 (dd, *J* = 12.4, 6.0 Hz, 2H), 1.57–1.47 (m, 2H), 0.84 (t, *J* = 7.3 Hz, 3H). ^13^C-NMR (100 MHz, DMSO-*d*_6_, ppm) δ 165.72, 163.14, 156.20, 152.53, 150.36(2,C), 149.53, 141.89, 135.72, 134.28(2,C), 132.86, 132.02, 130.02(2,C), 127.71, 122.60(2,C), 121.29(2,C), 114.17(2,C), 108.96, 40.57, 22.33, 11.24. TOF MS ES+ (*m/z*): [M + H]^+^, calcd for C_26_H_20_F_6_N_6_O_3_: 579.1580, found, 579.1573.

*4-(4-(1-(3,4-difluorophenyl)-5-(trifluoromethyl)-1H-1,2,3-triazole-4-carboxamido)phenoxy)-N-propylpicolinamide* (**B6**) Light yellow solid; Yield: 85.0%; m.p.: 97.6–97.8 °C; ^1^H-NMR (400 MHz, DMSO-*d*_6_, ppm) δ 11.21 (s, 1H), 8.80 (s, 1H), 8.52 (d, *J* = 4.9 Hz, 1H), 8.12 (s, 1H), 7.98 (d, *J* = 6.9 Hz, 2H), 7.81 (d, *J* = 8.9 Hz, 1H), 7.74 (s, 1H), 7.41 (s, 1H), 7.27 (d, *J* = 6.9 Hz, 2H), 7.20 (d, *J* = 2.3 Hz, 1H), 3.26–3.19 (m, 2H), 1.52 (dd, *J* = 13.7, 6.8 Hz, 2H), 0.84 (t, *J* = 7.1 Hz, 3H). ^13^C-NMR (100 MHz, DMSO-*d*_6_, ppm) δ 165.73, 163.13, 156.48, 152.54, 150.36(2,C), 149.48, 142.09, 135.81, 131.59, 124.44, 122.46(2,C), 121.32(2,C), 118.63, 118.44, 117.06, 116.86, 114.19(2,C), 108.93, 40.58, 22.34, 11.26. TOF MS ES+ (*m/z*): [M + H]^+^, calcd for C_25_H_19_F_5_N_6_O_3_: 547.1517, found, 547.1517.

*4-(4-(1-(3-chloro-4-fluorophenyl)-5-(trifluoromethyl)-1H-1,2,3-triazole-4-carboxamido)phenoxy)-N-propylpicolinamide* (**B7**) White solid; Yield: 81.6%; m.p.: 153.6–153.9 °C; ^1^H-NMR (400 MHz, DMSO-*d*_6_, ppm) δ 11.23 (s, 1H), 8.80 (t, *J* = 5.9 Hz, 1H), 8.52 (d, *J* = 5.6 Hz, 1H), 7.99 (s, 1H), 7.97 (s, 1H), 7.79 (d, *J* = 4.0 Hz, 3H), 7.42 (s, 1H), 7.27 (d, *J* = 8.6 Hz, 2H), 7.18 (d, *J* = 3.8 Hz, 1H), 3.23 (dd, *J* = 13.3, 6.5 Hz, 2H), 1.57–1.47 (m, 2H), 0.84 (t, *J* = 7.3 Hz, 3H). ^13^C-NMR (100 MHz, DMSO-*d*_6_, ppm) δ 165.73, 163.15, 156.61, 152.52, 150.33(2,C), 149.45, 142.38, 136.20, 135.87, 134.06, 129.69(2,C), 128.18(2,C), 122.39(2,C), 121.29(2,C), 114.13(2,C), 108.99, 40.58, 22.34, 11.25. TOF MS ES+ (*m/z*): [M + H]^+^, calcd for C_25_H_19_ClF_4_N_6_O_3_: 563.1222, found, 563.1216.

*4-(4-(1-(2-chloro-5-(trifluoromethyl)phenyl)-5-(trifluoromethyl)-1H-1,2,3-triazole-4-carboxamido)phenoxy)-N-propylpicolinamide* (**B8**) White solid; Yield: 56.7%; m.p.: 143.3–143.7 °C; ^1^H-NMR (400 MHz, DMSO-*d*_6_, ppm) δ 11.30 (s, 1H), 8.81 (s, 1H), 8.65 (s, 1H), 8.53 (d, *J* = 4.9 Hz, 1H), 8.21 (d, *J* = 8.0 Hz, 1H), 8.15 (d, *J* = 7.9 Hz, 1H), 8.00 (d, *J* = 8.4 Hz, 2H), 7.42 (s, 1H), 7.27 (d, *J* = 8.5 Hz, 2H), 7.19 (d, *J* = 2.1 Hz, 1H), 3.22 (dd, *J* = 12.2, 6.0 Hz, 2H), 1.52 (dq, *J* = 13.9, 7.1 Hz, 2H), 0.84 (t, *J* = 7.1 Hz, 3H). ^13^C-NMR (100 MHz, DMSO-*d*_6_, ppm) δ 165.72, 163.14, 156.13, 152.54, 150.37(2,C), 149.56, 141.97, 135.71, 135.05, 133.58, 131.78(2,C), 131.49, 130.44, 127.03, 122.63(2,C), 121.28(2,C), 114.18(2,C), 108.96, 40.58, 22.33, 11.25. TOF MS ES+ (*m/z*): [M + H]^+^, calcd for C_26_H_19_ClF_6_N_6_O_3_: 613.1190, found, 613.1189.

*4-(4-(1-(4-chloro-3-(trifluoromethyl)phenyl)-5-(trifluoromethyl)-1H-1,2,3-triazole-4-carboxamido)phenoxy)-N-propylpicolinamide* (**B9**) Light yellow solid; Yield: 66.3%; m.p.: 155.5–155.8 °C; ^1^H-NMR (400 MHz, DMSO-*d*_6_, ppm) δ 11.21 (s, 1H), 8.81 (s, 1H), 8.53 (d, *J* = 5.3 Hz, 1H), 8.44 (s, 1H), 8.17 (d, *J* = 8.1 Hz, 1H), 8.10 (d, *J* = 8.4 Hz, 1H), 7.99 (d, *J* = 8.7 Hz, 2H), 7.41 (s, 1H), 7.27 (d, *J* = 8.6 Hz, 2H), 7.19 (d, *J* = 2.9 Hz, 1H), 3.22 (dd, *J* = 12.8, 6.2 Hz, 2H), 1.52 (dq, *J* = 14.8, 7.4 Hz, 2H), 0.84 (t, *J* = 7.3 Hz, 3H). TOF MS ES+ (*m/z*): [M + H]^+^, calcd for C_26_H_19_ClF_6_N_6_O_3_: 613.1190, found, 613.1188.

*4-(2-fluoro-4-(5-methyl-1-phenyl-1H-1,2,3-triazole-4-carboxamido)phenoxy)-N-propylpicolinamide* (**B10**) White solid; Yield: 76.8%; m.p.: 143.3–143.7 °C; ^1^H-NMR (400 MHz, DMSO-*d*_6_, ppm) δ 10.96 (s, 1H), 8.83 (s, 1H), 8.54 (d, *J* = 5.4 Hz, 1H), 8.11 (d, *J* = 13.2 Hz, 1H), 7.86 (d, *J* = 8.7 Hz, 1H), 7.67 (d, *J* = 2.6 Hz, 5H), 7.48–7.39 (m, 2H), 7.23 (d, *J* = 4.7 Hz, 1H), 3.23 (dd, *J* = 12.9, 6.3 Hz, 2H), 2.59 (s, 3H), 1.57–1.47 (m, 2H), 0.84 (t, *J* = 7.3 Hz, 3H). ^13^C-NMR (100 MHz, DMSO-*d*_6_, ppm) δ 165.66, 163.43, 160.15, 153.09, 150.87(2,C), 138.46, 138.29, 135.61, 130.48, 130.09(3,C), 125.84(3,C), 124.07, 117.64, 113.82, 109.37, 108.46, 41.02, 22.73, 11.67, 9.86. TOF MS ES+ (*m/z*): [M + H]^+^, calcd for C_25_H_23_FN_6_O_3_: 475.1894, found, 475.1894.

*4-(2-fluoro-4-(1-(4-fluorophenyl)-5-methyl-1H-1,2,3-triazole-4-carboxamido)phenoxy)-N-propylpicolinamide* (**B11**) White solid; Yield: 65.1%; m.p.: 143.5–143.8 °C; ^1^H-NMR (400 MHz, DMSO-*d*_6_, ppm) δ 10.95 (s, 1H), 8.82 (s, 1H), 8.54 (d, *J* = 5.1 Hz, 1H), 8.10 (d, *J* = 12.7 Hz, 1H), 7.86 (d, *J* = 8.6 Hz, 1H), 7.76 (s, 2H), 7.51 (t, *J* = 8.4 Hz, 2H), 7.44 (d, *J* = 12.9 Hz, 2H), 7.22 (d, *J* = 2.5 Hz, 1H), 3.22 (d, *J* = 6.0 Hz, 2H), 2.58 (s, 3H), 1.51 (dd, *J* = 13.9, 6.9 Hz, 2H), 0.84 (t, *J* = 7.2 Hz, 3H). ^13^C-NMR (100 MHz, DMSO-*d*_6_, ppm) δ 165.65, 164.20, 163.42, 161.74, 160.11, 153.08, 150.88, 138.75, 138.22, 131.99, 128.44, 128.35, 124.08, 117.65, 117.18(2,C), 116.95, 113.83, 109.60, 109.38, 108.44, 41.01, 22.73, 11.67, 9.78. TOF MS ES+ (*m/z*): [M + H]^+^, calcd for C_25_H_22_F_2_N_6_O_3_: 493.1800, found, 493.1797.

*4-(2-fluoro-4-(5-methyl-1-(2-(trifluoromethoxy)phenyl)-1H-1,2,3-triazole-4-carboxamido)phenoxy)-N-propylpicolinamide* (**B12**) Light yellow solid; Yield: 63.2%; m.p.: 132.6–132.9 °C; ^1^H-NMR (400 MHz, DMSO-*d*_6_, ppm) δ 10.99 (s, 1H), 8.82 (s, 1H), 8.55 (d, *J* = 5.6 Hz, 1H), 8.10 (d, *J* = 13.2 Hz, 1H), 7.88 (d, *J* = 7.5 Hz, 1H), 7.85 (d, *J* = 8.0 Hz, 2H), 7.79 (d, *J* = 8.4 Hz, 1H), 7.73 (t, *J* = 7.5 Hz, 1H), 7.46 (d, *J* = 9.0 Hz, 1H), 7.41 (d, *J* = 2.5 Hz, 1H), 7.23 (dd, *J* = 5.5, 2.6 Hz, 1H), 3.26–3.20 (m, 2H), 2.46 (s, 3H), 1.58–1.47 (m, 2H), 0.85 (t, *J* = 7.4 Hz, 3H). TOF MS ES+ (*m/z*): [M + H]^+^, calcd for C_26_H_22_F_4_N_6_O_4_: 559.1717, found, 559.1717.

*4-(4-(1-(4-chlorophenyl)-5-(trifluoromethyl)-1H-1,2,3-triazole-4-carboxamido)-2-fluorophenoxy)-N-propylpicolinamide* (**B13**) Light yellow solid; Yield: 76.3%; m.p.: 176.5–176.8 °C; ^1^H-NMR (400 MHz, DMSO-*d*_6_, ppm) δ 11.37 (s, 1H), 8.81 (s, 1H), 8.55 (d, *J* = 5.3 Hz, 1H), 8.06 (d, *J* = 12.8 Hz, 1H), 7.79 (d, *J* = 7.4 Hz, 5H), 7.48 (d, *J* = 8.9 Hz, 1H), 7.44 (s, 1H), 7.27–7.20 (m, 1H), 3.23 (d, *J* = 6.1 Hz, 2H), 1.52 (dd, *J* = 14.1, 7.1 Hz, 2H), 0.84 (t, *J* = 7.3 Hz, 3H). TOF MS ES+ (*m/z*): [M + H]^+^, calcd for C_25_H_19_ClF_4_N_6_O_3_: 563.1222, found, 563.1216.

*4-(2-fluoro-4-(5-(trifluoromethyl)-1-(2-(trifluoromethyl)phenyl)-1H-1,2,3-triazole-4-carboxamido)phenoxy)-N-propylpicolinamide* (**B14**) Light yellow solid; Yield: 79.6%; m.p.: 172.8–173.1 °C; ^1^H-NMR (400 MHz, DMSO-*d*_6_, ppm) δ 11.43 (s, 1H), 8.82 (s, 1H), 8.55 (d, *J* = 5.3 Hz, 1H), 8.12 (s, 1H), 8.09 (d, *J* = 4.7 Hz, 1H), 8.05 (s, 1H), 8.00 (d, *J* = 9.1 Hz, 2H), 7.84 (d, *J* = 8.3 Hz, 1H), 7.49 (d, *J* = 8.9 Hz, 1H), 7.44 (s, 1H), 7.24 (d, *J* = 2.5 Hz, 1H), 3.23 (d, *J* = 5.6 Hz, 2H), 1.55–1.49 (m, 2H), 0.84 (t, *J* = 7.2 Hz, 3H). ^13^C-NMR (100 MHz, DMSO-*d*_6_, ppm) δ 165.15, 163.01, 156.39, 152.70, 150.48(2,C), 141.52, 134.25(2,C), 134.18, 132.86, 132.00, 129.99(2,C), 127.65, 123.86(2,C), 117.59, 113.43(2,C), 109.61, 109.39, 108.12, 40.60, 22.31, 11.21. TOF MS ES+ (*m/z*): [M + H]^+^, calcd for C_26_H_22_F_4_N_6_O_3_: 543.1614, found, 543.1616.

*4-(4-(1-(3,4-difluorophenyl)-5-(trifluoromethyl)-1H-1,2,3-triazole-4-carboxamido)-2-fluorophenoxy)-N-propylpicolinamide* (**B15**) White solid; Yield: 55.4%; m.p.: 134.4–134.8 °C; ^1^H-NMR (400 MHz, DMSO-*d*_6_, ppm) δ 11.40 (s, 1H), 8.83 (s, 1H), 8.55 (d, *J* = 5.4 Hz, 1H), 8.13 (d, *J* = 9.0 Hz, 1H), 8.06 (d, *J* = 13.0 Hz, 1H), 7.82 (d, *J* = 7.0 Hz, 2H), 7.74 (s, 1H), 7.48 (t, *J* = 8.8 Hz, 1H), 7.42 (s, 1H), 7.25 (d, *J* = 2.0 Hz, 1H), 3.22 (d, *J* = 6.2 Hz, 2H), 1.52 (d, *J* = 7.1 Hz, 2H), 0.84 (t, *J* = 7.1 Hz, 3H). TOF MS ES+ (*m/z*): [M + H]^+^, calcd for C_26_H_19_F_7_N_6_O_3_: 597.1485, found, 597.1486.

*4-(4-(1-(3-chloro-4-fluorophenyl)-5-(trifluoromethyl)-1H-1,2,3-triazole-4-carboxamido)-2-fluorophenoxy)-N-propylpicolinamide* (**B16**) White solid; Yield: 59.7%; m.p.: 163.3–163.5 °C; ^1^H-NMR (400 MHz, DMSO-*d*_6_, ppm) δ 11.39 (s, 1H), 8.82 (t, *J* = 5.8 Hz, 1H), 8.56 (d, *J* = 5.6 Hz, 1H), 8.23 (d, *J* = 4.8 Hz, 1H), 8.06 (d, *J* = 12.9 Hz, 1H), 7.88 (d, *J* = 8.5 Hz, 1H), 7.83 (d, *J* = 8.8 Hz, 1H), 7.77 (t, *J* = 8.9 Hz, 1H), 7.48 (t, *J* = 9.0 Hz, 1H), 7.42 (s, 1H), 7.25 (d, *J* = 5.4 Hz, 1H), 3.23 (dd, *J* = 13.3, 6.5 Hz, 2H), 1.53 (dt, *J* = 14.5, 7.4 Hz, 2H), 0.85 (t, *J* = 7.4 Hz, 3H). TOF MS ES+ (*m/z*): [M + H]^+^, calcd for C_25_H_18_ClF_5_N_6_O_3_: 581.1127, found, 581.1128.

4-(4-(1-(2-chloro-5-(trifluoromethyl)phenyl)-5-(trifluoromethyl)-1H-1,2,3-triazole-4-carboxamido)-2-fluorophenoxy)-N-propylpicolinamide (**B17**) White solid; Yield: 64.5%; m.p.: 154.1–154.4 °C; ^1^H-NMR (400 MHz, DMSO-d_6_, ppm) δ 11.48 (s, 1H), 8.83 (s, 1H), 8.65 (s, 1H), 8.55 (d, *J* = 5.0 Hz, 1H), 8.21 (d, *J* = 7.8 Hz, 1H), 8.15 (d, *J* = 8.2 Hz, 1H), 8.07 (d, *J* = 12.9 Hz, 1H), 7.84 (d, *J* = 8.6 Hz, 1H), 7.49 (t, *J* = 8.8 Hz, 1H), 7.43 (s, 1H), 7.25 (d, *J* = 1.4 Hz, 1H), 3.23 (d, *J* = 6.0 Hz, 2H), 1.52 (dd, *J* = 13.9, 6.9 Hz, 2H), 0.84 (t, *J* = 7.3 Hz, 3H). TOF MS ES+ (m/z): [M + H]^+^, calcd for C_26_H_18_ClF_7_N_6_O_3_: 631.1089, found, 631.1089.

4-(4-(1-(4-chloro-3-(trifluoromethyl)phenyl)-5-(trifluoromethyl)-1H-1,2,3-triazole-4-carboxamido)-2-fluorophenoxy)-N-propylpicolinamide (**B18**) White solid; Yield: 55.9%; m.p.: 153.4–153.7 °C; ^1^H-NMR (400 MHz, DMSO-d_6_, ppm) δ 11.41 (s, 1H), 8.83 (s, 1H), 8.56 (d, *J* = 5.3 Hz, 1H), 8.45 (s, 1H), 8.18 (d, *J* = 8.4 Hz, 1H), 8.12–8.01 (m, 2H), 7.84 (d, *J* = 8.4 Hz, 1H), 7.48 (t, *J* = 8.9 Hz, 1H), 7.42 (s, 1H), 7.25 (s, 1H), 3.26–3.19 (m, 2H), 1.52 (dq, *J* = 14.4, 7.0 Hz, 2H), 0.84 (t, *J* = 7.2 Hz, 3H). TOF MS ES+ (m/z): [M + H]^+^, calcd for C_26_H_18_ClF_7_N_6_O_3_: 631.1089, found, 631.1096.

*N-(3-fluoro-4-((2-(pyrrolidine-1-carbonyl)pyridin-4-yl)oxy)phenyl)-5-methyl-1-phenyl-1H-1,2,3-triazole-4-carboxamide* (**B19**) Light yellow solid; Yield: 63.2%; m.p.: 169.7–170.0 °C; ^1^H-NMR (400 MHz, DMSO-*d*_6_, ppm) δ 10.94 (s, 1H), 8.50 (d, *J* = 4.4 Hz, 1H), 8.09 (d, *J* = 12.9 Hz, 1H), 7.84 (d, *J* = 8.4 Hz, 1H), 7.65 (d, *J* = 14.4 Hz, 5H), 7.42 (t, *J* = 8.6 Hz, 1H), 7.13 (d, *J* = 14.2 Hz, 2H), 3.59 (s, 2H), 3.45 (s, 2H), 2.59 (s, 3H), 1.82 (s, 4H). TOF MS ES+ (*m/z*): [M + H]^+^, calcd for C_26_H_23_FN_6_O_3_: 487.1897, found, 487.1902.

*N-(3-fluoro-4-((2-(pyrrolidine-1-carbonyl)pyridin-4-yl)oxy)phenyl)-1-(4-fluorophenyl)-5-methyl-1H-1,2,3-triazole-4-carboxamide* (**B20**) Light yellow solid; Yield: 66.4%; m.p.: 164.3–164.7 °C; ^1^H-NMR (400 MHz, DMSO-*d*_6_, ppm) δ 10.94 (s, 1H), 8.51 (d, *J* = 5.3 Hz, 1H), 8.09 (d, *J* = 13.2 Hz, 1H), 7.84 (d, *J* = 8.4 Hz, 1H), 7.75 (d, *J* = 3.2 Hz, 2H), 7.51 (t, *J* = 8.4 Hz, 2H), 7.42 (t, *J* = 8.9 Hz, 1H), 7.18 (s, 1H), 7.12 (d, *J* = 4.6 Hz, 1H), 3.59 (s, 2H), 3.45 (s, 2H), 2.57 (s, 3H), 1.81 (s, 4H). TOF MS ES+ (*m/z*): [M + H]^+^, calcd for C_26_H_22_F_2_N_6_O_3_: 505.1800, found, 505.1807.

*N-(3-fluoro-4-((2-(pyrrolidine-1-carbonyl)pyridin-4-yl)oxy)phenyl)-5-methyl-1-(2-(trifluoromethoxy)phenyl)-1H-1,2,3-triazole-4-carboxamide* (**B21**) White solid; Yield: 66.5%; m.p.: 176.3–176.6 °C; ^1^H-NMR (400 MHz, DMSO-d_6_, ppm) δ 11.00 (s, 1H), 8.51 (d, *J* = 4.9 Hz, 1H), 8.09 (d, *J* = 12.8 Hz, 1H), 7.89 (d, *J* = 7.3 Hz, 1H), 7.85 (d, *J* = 8.0 Hz, 2H), 7.79 (d, *J* = 8.2 Hz, 1H), 7.75–7.70 (m, 1H), 7.43 (t, *J* = 8.6 Hz, 1H), 7.16 (s, 1H), 7.11 (d, *J* = 1.9 Hz, 1H), 3.60 (s, 2H), 3.46 (s, 2H), 2.46 (s, 3H), 1.82 (s, 4H). TOF MS ES+ (m/z): [M + H]^+^, calcd for C_27_H_22_F_4_N_6_O_4_: 571.1717, found, 571.1722.

*4-(2-fluoro-4-(5-methyl-1-phenyl-1H-1,2,3-triazole-4-carboxamido)phenoxy)-N-(2-(thiophen-2-yl)ethyl)picolinamide* (**B22**) Light yellow solid; Yield: 61.9%; m.p.: 124.6–125.0 °C; ^1^H NMR (400 MHz, DMSO-*d*_6_, ppm) δ 10.97 (s, 1H), 8.98 (t, *J* = 5.9 Hz, 1H), 8.55 (d, *J* = 5.6 Hz, 1H), 8.11 (d, *J* = 13.2 Hz, 1H), 7.86 (d, *J* = 8.8 Hz, 1H), 7.67 (d, *J* = 2.3 Hz, 5H), 7.48–7.41 (m, 2H), 7.32 (d, *J* = 4.9 Hz, 1H), 7.24 (d, *J* = 3.0 Hz, 1H), 6.96-6.91 (m, 1H), 6.89 (d, *J* = 2.6 Hz, 1H), 3.54 (dd, *J* = 13.4, 6.8 Hz, 2H), 3.06 (t, *J* = 7.1 Hz, 2H), 2.59 (s, 3H). TOF MS ES+ (*m/z*): [M + H]^+^, calcd for C_28_H_23_FN_6_O_3_S: 543.1614, found, 543.1616.

*4-(2-fluoro-4-(1-(4-fluorophenyl)-5-methyl-1H-1,2,3-triazole-4-carboxamido)phenoxy)-N-(2-(thiophen-2-yl)ethyl)picolinamide* (**B23**) White solid; Yield: 51.7%; m.p.: 110.4–110.8 °C; ^1^H-NMR (400 MHz, DMSO-*d*_6_, ppm) δ 10.99 (s, 1H), 9.00 (t, *J* = 5.7 Hz, 1H), 8.57 (d, *J* = 5.6 Hz, 1H), 8.13 (d, *J* = 13.2 Hz, 1H), 7.88 (d, *J* = 8.8 Hz, 1H), 7.79 (d, *J* = 4.8 Hz, 1H), 7.77 (d, *J* = 4.8 Hz, 1H), 7.54 (t, *J* = 8.7 Hz, 2H), 7.47 (d, *J* = 9.1 Hz, 1H), 7.43 (d, *J* = 2.1 Hz, 1H), 7.34 (d, *J* = 4.6 Hz, 1H), 7.26 (d, *J* = 2.4 Hz, 1H), 6.98–6.94 (m, 1H), 6.91 (s, 1H), 3.55 (dd, *J* = 13.3, 6.8 Hz, 2H), 3.08 (t, *J* = 7.1 Hz, 2H), 2.60 (s, 3H). TOF MS ES+ (*m/z*): [M + H]^+^, calcd for C_28_H_22_F_2_N_6_O_3_S: 561.1520, found, 561.1529.

*4-(2-fluoro-4-(5-methyl-1-(2-(trifluoromethoxy)phenyl)-1H-1,2,3-triazole-4-carboxamido)phenoxy)-N-(2-(thiophen-2-yl)ethyl)picolinamide* (**B24**) White solid; Yield: 56.6%; m.p.: 103.4–103.7 °C; ^1^H-NMR (400 MHz, DMSO-d_6_, ppm) δ 10.98 (s, 1H), 8.94 (s, 1H), 8.55 (d, *J* = 5.3 Hz, 1H), 8.10 (d, *J* = 13.2 Hz, 1H), 7.88 (d, *J* = 6.4 Hz, 2H), 7.84 (d, *J* = 7.0 Hz, 1H), 7.78 (d, *J* = 8.1 Hz, 1H), 7.72 (t, *J* = 7.4 Hz, 1H), 7.44 (t, *J* = 8.8 Hz, 2H), 7.31 (d, *J* = 4.1 Hz, 1H), 7.23 (d, *J* = 2.3 Hz, 1H), 6.96–6.92 (m, 1H), 6.90 (s, 1H), 3.55 (d, *J* = 6.2 Hz, 2H), 3.07 (t, *J* = 6.9 Hz, 2H), 2.47 (s, 3H). TOF MS ES+ (m/z): [M + H]^+^, calcd for C_29_H_22_F_4_N_6_O_4_S: 627.1437, found, 627.1439.

*4-(2-fluoro-4-(5-methyl-1-phenyl-1H-1,2,3-triazole-4-carboxamido)phenoxy)-N-(3-morpholinopropyl)picolinamide* (**B25**) White solid; Yield: 42.6%; m.p.: 108.3–108.6 °C; ^1^H-NMR (400 MHz, DMSO-*d*_6_, ppm) δ 10.67 (s, 1H), 9.05 (s, 1H), 8.53 (s, 1H), 8.03 (d, *J* = 7.5 Hz, 2H), 7.67 (s, 5H), 7.45 (s, 1H), 7.23 (d, *J* = 7.6 Hz, 1H), 7.17 (s, 1H), 3.64 (s, 4H), 3.36 (s, 2H), 2.60 (s, 3H), 2.44 (s, 6H), 1.72 (s, 2H). ^13^C-NMR (100 MHz, DMSO-*d*_6_, ppm) δ 166.25, 163.53, 159.92, 152.85, 150.72, 149.29, 138.56, 138.11, 136.82, 135.67, 130.44, 130.10(2,C), 125.83(2,C), 122.70(2,C), 121.51(2,C), 114.53, 109.30, 66.25(2,C), 56.81, 53.53(2,C), 38.36, 25.59, 9.85. TOF MS ES+ (*m/z*): [M + H]^+^, calcd for C_29_H_30_FN_7_O_4_: 560.2421, found, 560.2416.

*4-(2-fluoro-4-(1-(4-fluorophenyl)-5-methyl-1H-1,2,3-triazole-4-carboxamido)phenoxy)-N-(3-morpholinopropyl)picolinamide* (**B26**) White solid; Yield: 39.3%; m.p.: 108.5–108.9 °C; ^1^H-NMR (400 MHz, DMSO-*d*_6_, ppm) δ 10.73 (s, 1H), 9.10 (s, 1H), 8.52 (d, *J* = 3.7 Hz, 1H), 8.03 (d, *J* = 7.9 Hz, 2H), 7.76 (s, 2H), 7.52 (d, *J* = 6.9 Hz, 2H), 7.43 (s, 1H), 7.23 (d, *J* = 7.9 Hz, 2H), 3.61 (s, 4H), 3.35 (s, 2H), 2.59 (s, 3H), 2.34 (s, 6H), 1.68 (s, 2H). ^13^C-NMR (100 MHz, DMSO-*d*_6_, ppm) δ 166.25, 163.48, 159.87, 152.90, 150.69, 149.33, 138.38, 136.78, 132.06, 128.39, 128.30, 122.71(2,C), 121.46(2,C), 117.16(2,C), 116.93(2,C), 114.50, 109.32, 66.54(2,C), 57.08, 53.77(2,C), 38.55, 25.80, 9.74. TOF MS ES+ (*m/z*): [M + H]^+^, calcd for C_29_H_29_F_2_N_7_O_4_: 578.2327, found, 578.2347.

*4-(2-fluoro-4-(5-methyl-1-(2-(trifluoromethoxy)phenyl)-1H-1,2,3-triazole-4-carboxamido)phenoxy)-N-(3-morpholinopropyl)picolinamide* (**B27**) White solid; Yield: 41.0%; m.p.: 97.6–97.9 °C; ^1^H-NMR (400 MHz, DMSO-d_6_, ppm) δ 10.78 (s, 1H), 9.05 (s, 1H), 8.54 (s, 1H), 7.99 (d, *J* = 5.2 Hz, 2H), 7.86 (d, *J* = 3.8 Hz, 2H), 7.79 (s, 1H), 7.73 (s, 1H), 7.41 (s, 1H), 7.23 (d, *J* = 6.1 Hz, 2H), 3.91 (d, *J* = 8.3 Hz, 2H), 3.85 (d, *J* = 11.1 Hz, 2H), 3.36 (d, *J* = 5.0 Hz, 2H), 3.06 (s, 4H), 2.45 (s, 2H), 1.98 (s, 2H). TOF MS ES+ (m/z): [M + H]^+^, calcd for C_30_H_29_F_4_N_7_O_5_: 643.1059, found, 643.1094.

^1^H-NMR spectra of representative target compounds (**B1**, **B6**, **B12**, **B13**, **B20**, **B25**, and **B26**), ^1^^3^C-NMR spectra of representative target compounds (**B6**, **B7**, **B10**, **B11**, **B14**, **B25**, and **B26**), and TOF-MS spectra of representative target compounds (**B13**, **B17**, **B19**, **B21**, **B22**, **B25**, and **B26**) can be seen in the Appendix A.

### 3.2. Antitumority Assay

The antitumor activities of target compounds were determined by the MTT method using Golvatinib as a positive control. All cancer cell lines (A549, HeLa, and MCF-7) were cultured with Dulbecco Modified Eagle Medium (DMEM) or Roswell Park Memorial Institute (1640) containing 10% fetal bovine serum and 0.1% penicillin–streptomycin, under ambient conditions of 5% CO_2_ and 37 °C. Cells were digested with an appropriate amount of Trypsin–EDTA solution to obtain the cell suspension, which was diluted with medium and inoculated into 96-well plates at 5*10^4^ cells per well. After cells were incubated for 24 h, the target compounds diluted by medium to suitable concentrations were added into 96-well plates, and the cells were cultured continue for 72 h. Next, the medium was removed, and then thiazolyl blue tetrazolium bromide (MTT) was added to each well to treat cells for 3.5 h. Ultimately, dimethyl sulfoxide (DMSO) was added to each well after removal of MTT and the absorbance values were measured with the ELISA (enzyme-linked immunosorbent assay) reader. All antitumor activities were tested for three times. The IC_50_ values were the average of three measurements and were calculated using the Bacus Laboratories Incorporated Slide Scanner (Bliss) software.

### 3.3. Dose-Dependent Test

The dose-dependent effect of compound **B26** on A549 cells was tested by the MTT method using Golvatinib as a positive control. The experimental procedure was identical to that of the cytotoxic activity, wherein the concentration of the test compound was configured to be seven and the cytotoxic activity was five. Seven different concentrations of compound **B26** treated on A549 cells to obtain the corresponding inhibition rates. Experimental data was obtained by Orange (2018 64 bit) software based on the inhibition rate.

### 3.4. Cell Morphology Studies

Cell morphology studies aimed to explore the morphological changes of A549 cells with and without treatment with compound **B26**, and the cell morphology was visualized by acridine orange (AO) staining. The culture environment of A549 cells was consistent with that in the antitumor activity experiments. Cells were digested with Trypsin–EDTA solution to get the cell suspension, which was diluted with 1640 medium and inoculated into a 24-well plate at 2*10^4^ cells per well. After incubating for 12 h, compound **B26** diluted to suitable concentrations by 1640 medium was added into a 24-well plate, and cell culture was continued for 12 h. Then, the 1640 medium in the well plate was removed and every well were washed three times with phosphate buffer saline (PBS). After washing with PBS three times, A549 cells were treated with AO for 15 min. A549 cells were washed three times with PBS again, and the cell morphology distribution was observed directly by a fluorescence microscope. Ultimately, the picture is exported via computer.

### 3.5. Cell Cycle Study

A549 cells were seeded into six-well plates at 1*10^5^ cells per well and incubated for 24 h. Then, the medium containing DMEM (control, without compound) or compound **B26** with different concentrations was added to the culture plate, and A549 cells were further cultured for 24 h. Then, A549 cells were collected into a centrifuge tube and fixed with a small amount of ice-cold 70% ethanol at 4 °C for 6–12 h. Cells washed three times with PBS were incubated with Rnase (1 mg/mL, diluted with PBS) for 30 min at room temperature. Finally, propidium iodide (PI) was added to staining without light for 30 min at room temperature, and the DNA content was measured by flow cytometry within 1 h. Experimental data was obtained by Modify software.

### 3.6. c-Met Kinase Assay

The kinase assay was implemented through Mobility shift assay. The Mixture the configured available Brij-35 (0.0015%) and 50 mM HEPES (pH = 7.5) to get a kinase buffer base. Then, compounds **B25**, **B26**, and **B27** were configured to five concentrations using dimethyl sulfoxide. The kinase buffer containing the compounds or dimethyl sulfoxide was added to 96-well plates and mixed on the shaker for fifteen minutes. Next, the solution was transferred in duplicate from a 96-well plate to a 384-well plate, and then an enzyme solution (c-Met kinase mixed with kinase buffer) was added to each well. After incubation of the 384-well plate for 10 min at room temperature, a 2.5*peptide solution (formed by the addition of FAM-labeled peptide and ATP to the kinase base buffer) was added to each well. After incubation at 28 °C for a certain period of time, a stop buffer was added to each well for stop the reaction. The inhibition value is obtained by converting the conversion data on the caliper. The formula is that percent inhibition = (max − conversion) / (max − min) * 100. Among them, “max” stands for DMSO control, “min” stands for low control. Finally, the IC_50_ values were obtained by fitting the inhibition rate data using the XLFit excel add-in version software.

### 3.7. Molecular Docking Study

All the molecular docking simulations were performed by the AutoDock 4.2 software. The crystal structure of c-Met (PDB code: 3LQ8) used in the docking was downloaded from http://www.rcsb.org/. The preparation process of the protein for docking mainly involves the addition of hydrogen atoms and charges, elimination of unrelated water molecules, immobilization of exact residues, and removal of endogenous ligands (Foretinib), etc. The preparation process of the molecules for docking mainly includes the addition of hydrogen atoms and charges. Then the prepared molecules (target compounds) were docked to the certified binding site of c-Met protein. Subsequently, the genetic algorithm was used for energy optimization. All the docking results were modified and processed by PyMOL 1.8.x software (https://pymol.org).

## 4. Conclusions

In conclusion, a series of 4-(pyridin-4-yloxy)benzamide derivatives bearing a triazole fragment were designed and synthesized. In addition, we evaluated them for antitumor activity against three cancer cell lines and c-Met kinase activity (only for compounds **B25**–**B27**) in vitro. The pharmacological results indicated that most compounds showed moderate antitumor activity against the three cancer cell lines. In particular, compound **B26** showed excellent inhibitory activity with IC_50_ values of 3.22, 4.33, and 5.82 μM against A549, HeLa, and MCF-7 cell lines, which were more potent than Golvatinib, respectively. The SARs study indicated that the introduction of a morpholino group in the hydrophilic region was more favorable than an alkane chain in terms of antitumor activity, and a single electron-withdrawing substituent (such as a fluorine atom) on the terminal phenyl ring improved the inhibitory activity of the target compounds. Further studies will be carried out in the near future.

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
