# Peer review of "Design, Synthesis, and Biological Evaluation of Pyridineamide Derivatives Containing a 1,2,3-Triazole Fragment as Type II c-Met Inhibitors"

_molecules, 2019, doi:10.3390/molecules25010010_

Round 1

Reviewer 1 Report

The manuscript describes the synthesis of novel 4-(pyridin-4-yloxy)benzamide derivatives containing 1,2,3-triazole fragment and their antiproliferative activity against cancer cells. The most promising compound B26 was further studied to determine the effects on cell cycle and c-Met inhibitory activity. Molecular docking studies were performed too.

Overall the manuscript would be of interest to the readers, however I found some errors and unclear points, therefore before publication substantial changes should be performed.

English language is not always correct. For example, see the sentences evidenced in yellow in the attached file. Please, check all the manuscript.

Paragraph 2.1. Chemistry

In my opinion, describing the synthesis of compounds already reported in the literature is redundant. Therefore, Scheme 1 should be eliminated and Scheme 2 reduced to the last step. Consequently, also the text of the Chemistry section should be reduced, together with the Chemistry paragraph of the Experimental Section (3.1 Chemistry).

Paragraph 2.5. c-Met Enzyme Assay of Compounds B25-B27 and Staurosporine.

The inhibition rate of the studied compounds is much lower than that of Staurosporine. Please, comment.

Moreover, in my opinion, it would be interesting to report Golvatinib c-Met inhibition activity, in order to compare it to B26 inhibitory data. This point would be important also to comment the molecular docking data, which concern Golvatinib and B26.

Author Response

Dear Reviewer 1,

Thanks for your comments on our work. Our manuscript Molecules-661230 was revised according to your comments. All revisions have been highlighted using the "Track Changes" function in Microsoft Word. The itemized response to each comment was attached.

The manuscript describes the synthesis of novel 4-(pyridin-4-yloxy)benzamide derivatives containing 1,2,3-triazole fragment and their antiproliferative activity against cancer cells. The most promising compound B26 was further studied to determine the effects on cell cycle and c-Met inhibitory activity. Molecular docking studies were performed too. Overall the manuscript would be of interest to the readers, however I found some errors and unclear points, therefore before publication substantial changes should be performed.

Point 1: English language is not always correct. For example, see the sentences evidenced in yellow in the attached file. Please, check all the manuscript. 

Response 1: Thank you for your kindly reminding. With the help of several English-speaking colleagues, we checked all the manuscript carefully and tried our best to correct those problems to make language concise, reasonable and smooth. All of the corrections in the revised manuscript were marked using the "Track Changes" function in Microsoft Word.

Point 2: Paragraph 2.1. Chemistry

In my opinion, describing the synthesis of compounds already reported in the literature is redundant. Therefore, Scheme 1 should be eliminated and Scheme 2 reduced to the last step. Consequently, also the text of the Chemistry section should be reduced, together with the Chemistry paragraph of the Experimental Section (3.1 Chemistry). English language is not always correct. For example, see the sentences evidenced in yellow in the attached file. Please, check all the manuscript.

Response 2: Thanks for your constructive suggestion. I'm very sorry about that the position of the references inserted may cause some misunderstanding to you, so I think a specific explanation is necessary. The references cited [15-17] are only enlightening for some in the synthesis steps of Scheme 1, which was explored by us independently, and so the synthetic route of 12a-12e in Scheme 1 has not been reported so far. Therefore, the whole synthesis process and methods in Scheme 1 was reserved. The preparation of compounds 13a-13i has been reported in our previous work [15], so the synthesis method of 13a-13i and ‘Scheme 2’ has been deleted. According to your suggestion, we kept the last step of Scheme 2. Finally, we combined Scheme 1 and the last step of Scheme 2 to a new route in the revised manuscript.

Point 3: Paragraph 2.5. c-Met Enzyme Assay of Compounds B25-B27 and Staurosporine. The inhibition rate of the studied compounds is much lower than that of Staurosporine. Please, comment. Moreover, in my opinion, it would be interesting to report Golvatinib c-Met inhibition activity, in order to compare it to B26 inhibitory data. This point would be important also to comment the molecular docking data, which concern Golvatinib and B26.

Response 3: Thank you for your constructive suggestion. Staurosporine (provided by the company of enzyme activity test) is only a control to ensure the reliability of the test data, so it is not necessary to compare the c-Met enzyme activity of compounds B25-B27 and Staurosporine. Our main purpose of c-Met enzyme activity test is to check whether the compounds have inhibition effect on c-Met. In our further study, we will compare the c-Met enzyme activity of B26 and Golvatinib. Thanks again for your advice.

Reviewer 2 Report

An interesting synthetic and biological study. The chemistry text describing the synthesis of 1,2,3-triazole containing 4-(pyridin-4-yloxy)benzamide derivatives should be rewritten in a manner that describes the chemical transformations taking place.

In Scheme 1 compound 6 should be a 4-chloropyrimidine-6-carboxylic acid.

IUPAC nomenclature should be given to names of compounds in the Experimental section.

Several English grammar mistakes should be corrected.

Author Response

Dear Reviewer 2,

Thanks for your comments on our work. Our manuscript Molecules-661230 was revised according to your comments. All revisions have been highlighted using the "Track Changes" function in Microsoft Word. The itemized response to each comment was attached.

An interesting synthetic and biological study.

Point 1: The chemistry text describing the synthesis of 1,2,3-triazole containing 4-(pyridin-4-yloxy)benzamide derivatives should be rewritten in a manner that describes the chemical transformations taking place. 

Response 1: Thank you very much for your kindly reminding. According to your suggestion, the chemistry text describing the synthesis of 1,2,3-triazole containing 4-(pyridin-4-yloxy)benzamide derivatives has been revised accordingly.

Point 2: In Scheme 1 compound 6 should be a 4-chloropyrimidine-6-carboxylic acid.

Response 2: Thank you very much for reading my manuscript carefully. Firstly, the picolinic acid was reacted with SOCl2 to obtain picolinoyl chloride, which was then converted into 4-chloropicolinoyl chloride with NaBr as a catalyst. So, compound 6 in Scheme 1 is 2-picolinic acid indeed, not 4-chloropyrimidine-6-carboxylic acid.

Point 3: IUPAC nomenclature should be given to names of compounds in the Experimental section.

Response 3: Thanks for your constructive suggestion. All compounds were named using IUPAC nomenclature in the Experimental section.

Point 4: Several English grammar mistakes should be corrected.

Response 4: Thank you for your kindly reminding. With the help of several English-speaking colleagues, we checked the manuscript carefully and tried our best to correct those problems to make language concise, reasonable and smooth. All of the corrections in the revised manuscript were marked using the "Track Changes" function in Microsoft Word.

Round 2

Reviewer 1 Report

I think the response to the third point “Paragraph 2.5. c-Met Enzyme Assay of Compounds B25-B27 and Staurosporine. The inhibition rate of the studied compounds is much lower than that of Staurosporine. Please, comment. Moreover, in my opinion, it would be interesting to report Golvatinib c-Met inhibition activity, in order to compare it to B26 inhibitory data. This point would be important also to comment the molecular docking data, which concern Golvatinib and B26. “is not adequate. However, if the Editor decides to accept the revised manuscript for publication, it is OK also for me.